# A minimalist approach to stereoselective glycosylation with unprotected donors

Kim Le Mai Hoang [iD] [1], Jing-xi He[2], Gábor Báti[1], Mary B. Chan-Park[2] & Xue-Wei Liu[1]

Mechanistic study of carbohydrate interactions in biological systems calls for the chemical synthesis of these complex structures. Owing to the specific stereo-configuration at each anomeric linkage and diversity in branching, significant breakthroughs in recent years have focused on either stereoselective glycosylation methods or facile assembly of glycan chains. Here, we introduce the unification approach that offers both stereoselective glycosidic bond formation and removal of protection/deprotection steps required for further elongation. Using dialkylboryl triflate as an in situ masking reagent, a wide array of glycosyl donors carrying one to three unprotected hydroxyl groups reacts with various glycosyl acceptors to furnish the desired products with good control over regioselectivity and stereoselectivity. This approach demonstrates the feasibility of straightforward access to important structural scaffolds for complex glycoconjugate synthesis.

[1] Division of Chemistry and Biological Chemistry, School of Physical and Mathematical Sciences, Nanyang Technological University, 21 Nanyang Link, Singapore 637371, Singapore. [2] School of Chemical and Biomedical Engineering, Nanyang Technological University, 62 Nanyang Drive, Singapore 637459, Singapore. Correspondence and requests for materials should be addressed to M.B.C.-P. (email: mbechan@ntu.edu.sg) or to X.-W.L. (email: xuewei@ntu.edu.sg)

The total synthesis of a seventy-seven nucleotide unit long DNA duplex, coding for the yeast alanine tRNA, by Khorana's group[1] in 1972 is regarded[2] as one of the "greatest *tour de force* organic and biochemists have yet achieved". This landmark accomplishment at the time prefaced the explosive growth of knowledge in biochemistry research for many decades and the halo effect accelerated many radical innovations in the synthesis of oligonucleotides, oligopeptides and oligosaccharides. The slower pace of breakthroughs in oligosaccharide synthesis is certainly not from limited interest into the field, but rather the inherently complex and diverse structures employed by biological systems[3, 4]. The remarkable density of

**Fig. 1** Strategic overview of chemical glycosylation methods. **a**, General profile of a glycosylation reaction. **b**, C-2 neighboring group participation for controlled 1,2-*cis* or 1,2-*trans* linkages. **c**, high β-mannosylation and high α-galactosylation employing ring-locked conformation. **d**, high *cis*- selectivity for aminoglycosyl and high α-sialylation with cyclic carbamate protection. **e**, an example of acceptor-controlled stereoselectivity. **f**, stereoselective and regioselective glycosylation with temporal boron-masking allows flexible switching between donors and acceptors and removal of protection/deprotection step (this work)

**Fig. 2** Boron-mediated glycosylation and their theoretical intermediates. **a**, IAD-like glycosylation with small alkyl groups on boron to form 1,2-*cis* product. **b**, 1,2-*trans* glycosylation with bulky alkyl groups

### Table 1 Optimization studies

| Entry[a] | Reagent 2 | T1 | Reagent 3 | T2 | Solvent | Yield (%)[b] | α/β[c] |
|---|---|---|---|---|---|---|---|
| 1 | MeOH | −20 °C | DMTSF | −20 °C | $CH_2Cl_2$ | **1b**, 60 | 1/1.5 |
| 2 | EtOH | −20 °C | DMTSF | −20 °C | $CH_2Cl_2$ | **1c**, 62 | 1/1.8 |
| 3 | *i*-PrOH | −20 °C | DMTSF | −20 °C | $CH_2Cl_2$ | **1d**, 59 | 1/2.5 |
| 4 | BuOH | −20 °C | DMTSF | −20 °C | $CH_2Cl_2$ | **1e**, 68 | 1/4.1 |
| 5 | BuOH | −20 °C | DMTSF | −20 °C | DMF | **1e**, 30 | β |
| 6 | BuOH | −20 °C | DMTSF | −20 °C | $CH_3CH_2NO_2$ | **1e**, 50 | 1/2 |
| 7 | BuOH | −20 °C | DMTSF | −20 °C | Toluene | **1e**, 40 | 1/6 |
| 8 | BuOH | −20 °C | DMTSF | −20 °C | $CH_3CN$ | **1e**, trace | 1/5 |
| 9 | BuOH | −40 °C | DMTSF | −40 °C | $CH_2Cl_2$ | **1e**, 73 | β |
| 10[d] | BuOH | −40 °C | NIS/TMSOTf | −40 °C | $CH_2Cl_2$ | **1e**, 89 | β |
| 11[e] | *p*-NO₂PhSCl | −60 °C | BuOH | −78 °C | $CH_2Cl_2$ | **1e**, 85 | β |
| 12[f] | Ph₂SO/Tf₂O | −60 °C | BuOH | −78 °C | $CH_2Cl_2$ | **1e**, 86 | β |

[a]Unless otherwise specified, all reactions were carried out with 1 equivalent of **1a**, 1.1 eq. of boron reagent, 1.2 eq. of ROH and 3 eq. of DMTSF in 2 mL of solvent for 12 h
[b]isolated yield
[c]determined by ¹H-NMR integration
[d]1.2 eq. of NIS and 0.2 eq. of TMSOTf were added
[e]3 eq. of AgOTf and 1.2 eq. of *p*-NO₂PhSCl were added
[f]1.3 eq. of Ph₂SO and 1.5 eq. of Tf₂O were added. n.d. not detected

information that carbohydrates can carry stemmed from their ability to form branching connections and the unique stereo-configuration between each sugar unit. Compounding the difficulty in isolating homogeneous carbohydrates from living cells is the "micro-heterogeneity", releasing many variants of the core oligosaccharide scaffold due to minute changes in the environment. At present, classic chemical or hybrid chemoenzymatic approach provide the most reliable access to homogenous form of glycans[5–9]. Figure 1 highlights the core strategies of a glycosylation reaction. Nucleophilic substitution at the anomeric center is the crucial bottleneck as clean $S_N2$ transformation is almost always accompanied by competing $S_N1$ due to stabilization of C1 cation by the endocyclic oxygen to generate oxocarbenium ion. Controlled outcome of this delicate equilibrium is notoriously difficult since it is easily influenced by many factors, including reactivity of glycosyl donors, temperature, solvents, other additives, etc. (Fig. 1a).

To date, specific requirements to deliver a high degree of stereoselectivity have emerged, such as employment of neighboring group participation at C-2 to deliver cis- or trans- selective products[10–13] (Fig. 1b), remote stereo-control with locked conformation[14–20] (Figs. 1c, d) or acceptor-controlled glycosylation[21, 22] (Fig. 1e). The successful answers to problems of stereoselective *O*-glycosylation came with drawbacks in the form of extensive preparative routes or usage of exotic chiral auxiliaries. Different approaches had been reported from other groups towards a protection-free glycosylation, such as glycosyl dithiocarbamates[23], oxathiane glycosyl donors[24], or boron-mediated glycosylation. The latter mainly focused on employing diol glycosyl acceptors as binding substrates[25–32].

In addition to development of better stereo-controlled methodology, notable effort has been directed towards designing a simpler and more efficient assembly of complex glycans from relatively small pool of building blocks. Major breakthroughs

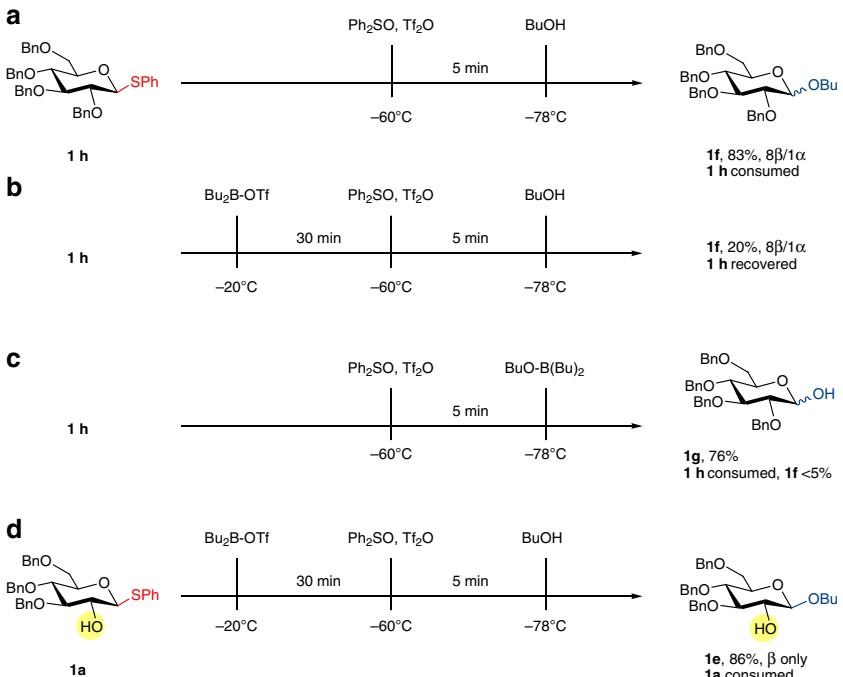

**Fig. 3** Control experiments. **a**, typical glycosylation under Ph$_2$SO/Tf$_2$O activation. **b**, inhibit activation in the absence of free hydroxyl group on donor. **c**, sluggish glycosylation with pre-masked butanol. **d**, masking of glycosyl donor's hydroxyl group successfully delivered desired product

included one-pot glycosylation, chemoenzymatic approach and automated synthesis on solid support. Based on early observation made by Paulsen[33] and Fraser-Reid[34], the concept of armed-disarmed glycosyl donors was further refined by Ley[35] and Wong[36] into a predictive way to quantify the relative reactivity between donors carrying different electron-withdrawing groups. Careful choice of protective groups and orthogonal activation in an one-pot setup allowed the synthesis of many complex oligosaccharides[37, 38], with the advantages of minimal isolation and purifications. Certain limits do exist, such as extensive protective groups tunings, possible self-condensation or crosslinking and unwieldy design when there are more than one branching in the glycan structure. Capitalizing on the knowledge and availability of current glycosyl transferases and bio-engineered glycosyl hydrolases, so-called glycosyl synthases[39], a hybrid chemoenzymatic approach saw practice in challenging modifications such as terminal α-2,3- and α-2,6-sialylation[40] or core α-fucosylation[41], library synthesis of asymmetrically branched, complex N-glycans[42, 43], or top-down trimming of high-mannose type N-glycans[44]. Predicaments to widespread application of this excellent strategy included feedback inhibition, commercial availability and desired regioselectivity. On the other hand, automated glycosylation[45] is at its nascent stage now but is fast gaining tractions among the community, with major improvements expected to come. The latest record-holder for the longest chemically prepared carbohydrate structure was a 92-mer polysaccharide arabinogalactan[46], whereas the former champion was a 30-mer repeating mannosides[47] using automated solid-phase oligosaccharide synthesis. Currently, the most resource-consuming stage is arguably the complete and selective deprotection after every coupling step.

In this work, we aim to provide a solution to the aforementioned challenges with a well-designed temporal group that binds to glycosyl donor in situ, prior to activation, to direct the stereo-outcome but is removed in the work-up phase (Fig. 1f). The result resembles a protection-free glycosylation strategy whereas one, two, or three free hydroxyl groups on donor can be present

without detrimental effects on reaction yield and selectivity. This is made possible through combination of temporal boron-protection, stoichiometric activation and sequential addition of glycosyl acceptors. Our approach has the general advantages of employing commonly used thioglycoside donors, flexibility in the number and position of free hydroxyl groups as well as wide compatibility with many activation methods.

## Results

**Optimizing and control experiments.** We set out to investigate the use of 2-OH glucosyl donor **1a** as the model substrate with various simple alcohol acceptors. Initially, we proposed that usage of different dialkylboryl triflates could have marked effects on the stereo-outcome of reaction. Smaller alkyl groups such as dimethyl may lead to formation of a tetravalent boron intermediate, which facilitate delivery of glycosyl acceptor from the same face to give 1,2-*cis* product (Fig. 2a). In principle, the mechanism should be similar to the well-studied intramolecular aglycon delivery (IAD)[48, 49]. A conceptually analogous model was proposed by Toshima[28, 29], starting from 1,2-anhydrosugars instead. On the other hand, larger alkyl groups such as di-*n*-butyl or 9-BBN should provide steric hindrance to force glycosyl acceptor to approach the anomeric center from opposite face, giving 1,2-*trans* product (Fig. 2b). This time, the boron complex was acting like a directing group through its spatial occupancy.

Optimizing masking effect of boron and several low temperature NMR experiments dismissed the first tetravalent model (Fig. 2a) and suggested the formation of the latter (Fig. 2b). In addition, the boron-masked O-2 exhibited an electron-withdrawing effect on H-2 proton as it was shifted downfield to 4.20 ppm from its previous position at 3.56 ppm (Δ = 0.64 ppm), see Supplementary Discussions. Furthermore, adding sequence of reagents was critical for consistent formation of desired product. The following protocol was established: An equimolar amount of dialkyl boron triflate was transferred to glycosyl donor **1a**, followed by addition of thiophilic activator and glycosyl acceptor.

**Table 2 Scope of single unprotected hydroxyl glucosyl donor**

| Entry[a] | Donor | Product (Yield[b] %, α/β[c]) | Entry[a] | Donor | Product (Yield[b] %, α/β[c]) |
|---|---|---|---|---|---|
| 1 | 1a | 1e, 86%, β only | 7 | 7a | 7e, 86%, 1α/8β |
| 2 | 2a | 2e, 83%, β only | 8 | 8a | 8e, 85%, β only |
| 3 | 3a | 3e, 72%, 1α/4.5β | 9 | 9a | 9e, 91%, β only |
| 4 | 4a | 4e, 88%, β only | 10 | 10a | 10e, 82%, 1α/2β |
| 5 | 5a | 5e, 87%, β only | 11 | 11a | 11e, 83%, β only |
| 6 | 6a | 6e, 81%, 1α/6β | 12 | 12a | 12e, 91%, β only |

[a]Unless otherwise specified, all reactions were carried out with 1 equivalent of donor, 1.1 eq. of Bu$_2$BOTf, 1.3 eq. of Ph$_2$SO, 3 eq. of TTBP, 1.5 eq. of Tf$_2$O (1 M in CH$_2$Cl$_2$), 1.2 equivalent of BuOH in 2 mL of solvent for 12 h
[b]Isolated yield
[c]determined by $^1$H-NMR integration

At −20 °C, there was little stereoselectivity when methanol was used as acceptor and dimethyl(methylthio) sulfonium tetrafluoroborate (DMTSF) as activator. Nevertheless, product **1b** was the only glucoside detected, confirming the free hydroxyl group on **1a** was successfully masked and no self-condensation occurred. On the other hand, we noticed an increase in β-product moving from methanol to ethanol, isopropanol and n-butanol (Table 1, entries 1–4), suggested that steric bulk of alkyl chain does play an appreciable role in stereo-outcome of reaction. Using *n*-butanol as the acceptor, survey of various solvents revealed CH$_2$Cl$_2$ afforded the best yield with moderate stereo-selectivity, amidst DMF, nitroethane, and toluene (Table 1, entries 5–7). No chemical conversion was observed when acetonitrile was used, and all starting materials were recovered (Table 1, entry 8). Lower reaction temperature to −40 °C resulted in pure β-**1e** (Table 1, entry 9). Within sensitivity limit of NMR instrument the α-product was undetected. It was very encouraging that many commonly used activation conditions, including DMTSF, NIS/TMSOTf, AgOTf/pNO$_2$PhSCl and Ph$_2$SO/Tf$_2$O were compatible

with the addition of boron reagent (Table 1, entries 9–12). Entries 1–4 were repeated with other dialkyl boron triflates such as diethyl, 9-BBN or dicyclohexyl but no appreciable change in selectivities were observed. Likewise, these dialkyl boron triflates delivered β-only **1e** under optimized conditions (entries 9–12).

A set of control experiments was performed to understand the probable mechanism. Typical glycosylation with fully protected glycosyl donor **1h** with Ph$_2$SO/Tf$_2$O yielded product **1f** as a mixture of anomers (Fig. 3a). When Bu$_2$BOTf was introduced to the mixture of **1h** and Ph$_2$SO, followed by drop-wise addition of Tf$_2$O and then butanol at −60 °C, very little amount of product was observed and most glycosyl donor was recovered (Fig. 3b). Premixing butanol and Bu$_2$BOTf to generate BuO-B(Bu)$_2$ likewise led to very little desired product, with major hydrolysis occurred (Fig. 3c). As expected, Ph$_2$SO/Tf$_2$O activating condition was not compatible with unprotected hydroxyl glycosyl donor[50, 51]. We reached the conclusion that without a free hydroxyl group to react, Bu$_2$BOTf inhibited the activation process and slowed down the reaction. In addition, hydroxyl group

**Table 3 Scope of multiple unprotected hydroxyl glucosyl donor**

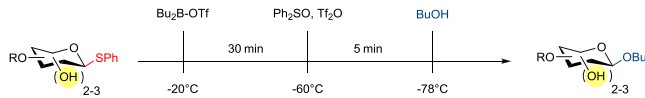

| Entry[a] | Donor | Product (Yield[b] %, α/β[c]) | Entry[a] | Donor | Product (Yield[b] %, α/β[c]) |
|---|---|---|---|---|---|
| 1 | 13a | 13e, 88%, β only | 7 | 19a | 19e, 78%, 1α/11β |
| 2 | 14a | 14e, 86%, β only | 8 | 20a | 20e, 79%, β only |
| 3 | 15a | 15e, 83%, 1α/10β | 9[d] | 21a | 21e, 79%, 1α/6β |
| 4 | 16a | 16e, 79%, 1α/1.1β | 10[d] | 22a | 22e, 81%, β only |
| 5 | 17a | 17e, 84%, 1α/20β | 11[d] | 23a | 23e, 80%, 1α/2.9β |
| 6 | 18a | 18e, 80%, 1α/2.5β | 12[d] | 24a | 24e, 80%, 1α/1.7β |

[a]Unless otherwise specified, all reactions were carried out with 1 equivalent of donor, 2.1 eq. of $Bu_2BOTf$, 1.3 eq. of $Ph_2SO$, 3 eq. of TTBP, 1.5 eq. of $Tf_2O$ (1 M in $CH_2Cl_2$), 1.2 equivalent of BuOH in 2 mL of solvent for 12 h
[b]isolated yield
[c]determined by [1]H-NMR integration
[d]3.1 eq. of $Bu_2BOTf$ was used

masked by $Bu_2BOTf$ were no longer able to act as acceptor. This would explain the necessity to mix donor with equimolar amount of boron reagent before thiophilic activators were introduced (Fig. 3d).

**Expanding the substrate scope**. With the optimized reaction conditions in hand, the scope of glucosyl donor carrying a single free hydroxyl group at different positions was first evaluated. As seen from Table 2, we observed a strong β-selectivity with 2-OH and 3-OH glucosyl donor. Interestingly, selectivity was reduced when there was an acetyl at adjacent position (Table 2, entry 3 and 6). [11]B-NMR experiments revealed an upfield shift to about −20 ppm compared to the chemical shift of −3 ppm with donor **1a**, suggesting the formation of a different tetravalent boron intermediate (see Supplementary Discussions). It was probable the acyl group formed a complex with boron center, thus attenuated the steric effect of the auxiliary.

With 4-OH glucosyl donor, selectivity with benzylated **7a** was very similar to control experiment with perbenzylated **1h**. Still, the masking effect of boron reagent could be used in combination with a stronger directing group, such as 2-O-benzoyl (Table 2,

entry 8). Notably, glucosyl donor **9a** having an inverted conformation $^1C_4$ resulted in intramolecular displacement of PMB group at O−6 to form 1,6-anhydroglucoside **9e** (Table 2, entry 9). If no boron was added and DMTSF was used as activating agent, a complicated mixture was observed and only trace amount of **9e** was isolated (<5%). With 6-OH glucosyl donor, significant amount of α-product was formed, raising α/β to1/2. We believed the β-facing of primary 6-OH was responsible for this rise in α-anomers. Again, presence of a directing group such as 2-O-benzoyl restored the selectivity and deliver the desired product in good yield (Table 2, entry 11). As expected, glucosyl donor **12a** with inverted conformation $^1C_4$ resulted in formation of 1,6-anhydroglucoside **12e** as the only product (Table 2, entry 12).

The substrate scope of glucosyl donors carrying more free hydroxyl groups was examined next. As shown in Table 3, the general trend from Table 2 was noticed. High β-selectivity was observed when there was either 2-OH or 3-OH available (Table 3, entries 1–3, 5) but was strongly reduced when 6-OH was also present (Table 3, entry 4 and 6). As expected, compound **20a** equipped with 2-O-benzoyl smoothly produced β-**20e** as the sole

**Table 4 Scope of disaccharides**

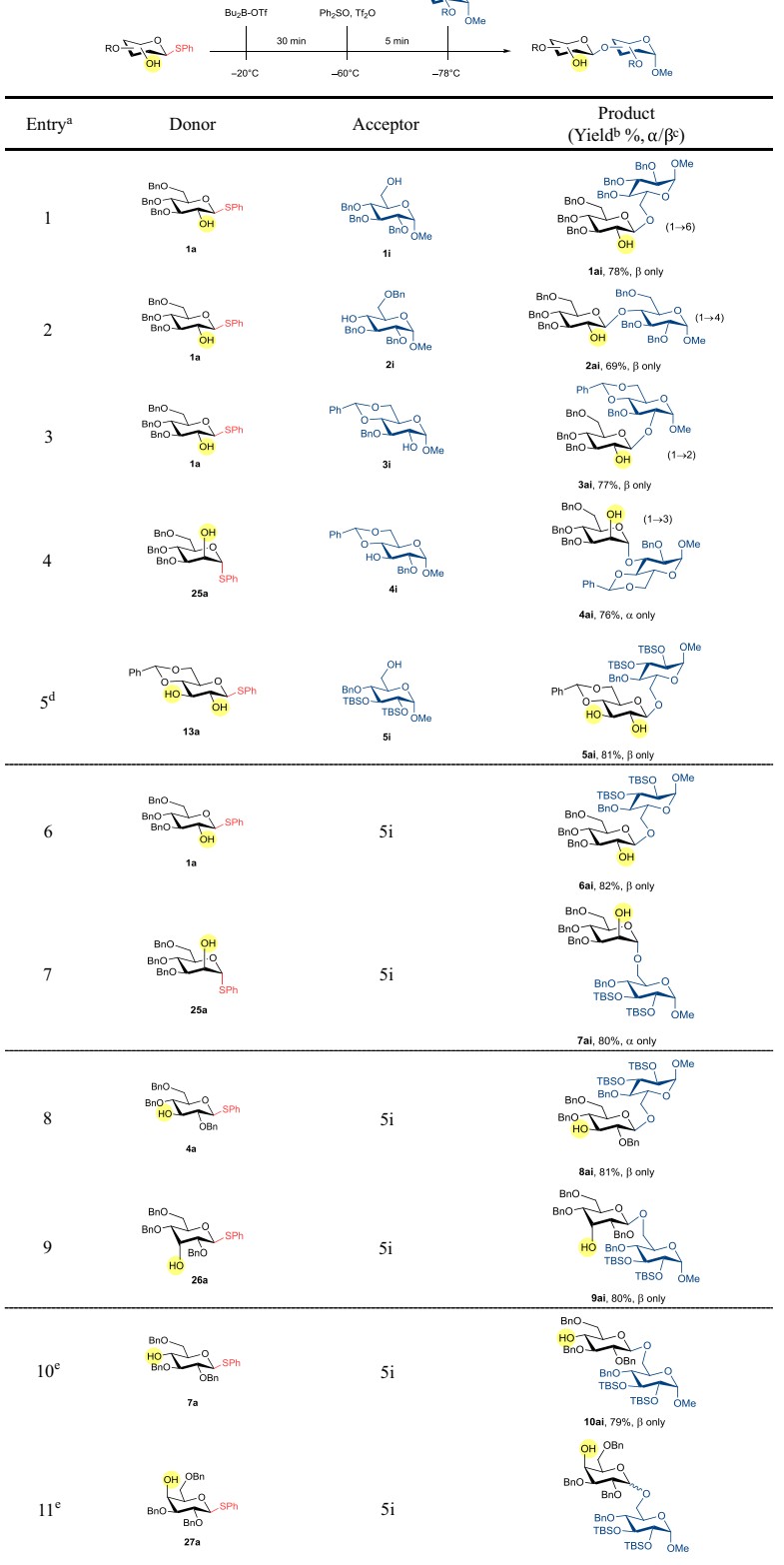

| Entry[a] | Donor | Acceptor | Product (Yield[b] %, α/β[c]) |
|---|---|---|---|
| 1 | **1a** | **1i** | **1ai**, 78%, β only (1→6) |
| 2 | **1a** | **2i** | **2ai**, 69%, β only (1→4) |
| 3 | **1a** | **3i** | **3ai**, 77%, β only (1→2) |
| 4 | **25a** | **4i** | **4ai**, 76%, α only (1→3) |
| 5[d] | **13a** | **5i** | **5ai**, 81%, β only |
| 6 | **1a** | **5i** | **6ai**, 82%, β only |
| 7 | **25a** | **5i** | **7ai**, 80%, α only |
| 8 | **4a** | **5i** | **8ai**, 81%, β only |
| 9 | **26a** | **5i** | **9ai**, 80%, β only |
| 10[e] | **7a** | **5i** | **10ai**, 79%, β only |
| 11[e] | **27a** | **5i** | **11ai**, 77%, 1α/2.5β |

[a]Unless otherwise specified, all reactions were carried out with 1 equivalent of donor, 1.1 eq. of Bu$_2$BOTf, 1.3 eq. of Ph$_2$SO, 3 eq. of TTBP, 1.5 eq. of Tf$_2$O (1 M in CH$_2$Cl$_2$), 1.2 eq. of acceptor in 2 mL of solvent for 12 h
[b]isolated yield
[c]determined by $^1$H-NMR integration
[d]2.1 eq. of Bu$_2$BOTf was used
[e]1.1 eq. of Cy$_2$BOTf was used

**Table 5 Glycosylation with dual role donor/acceptor**

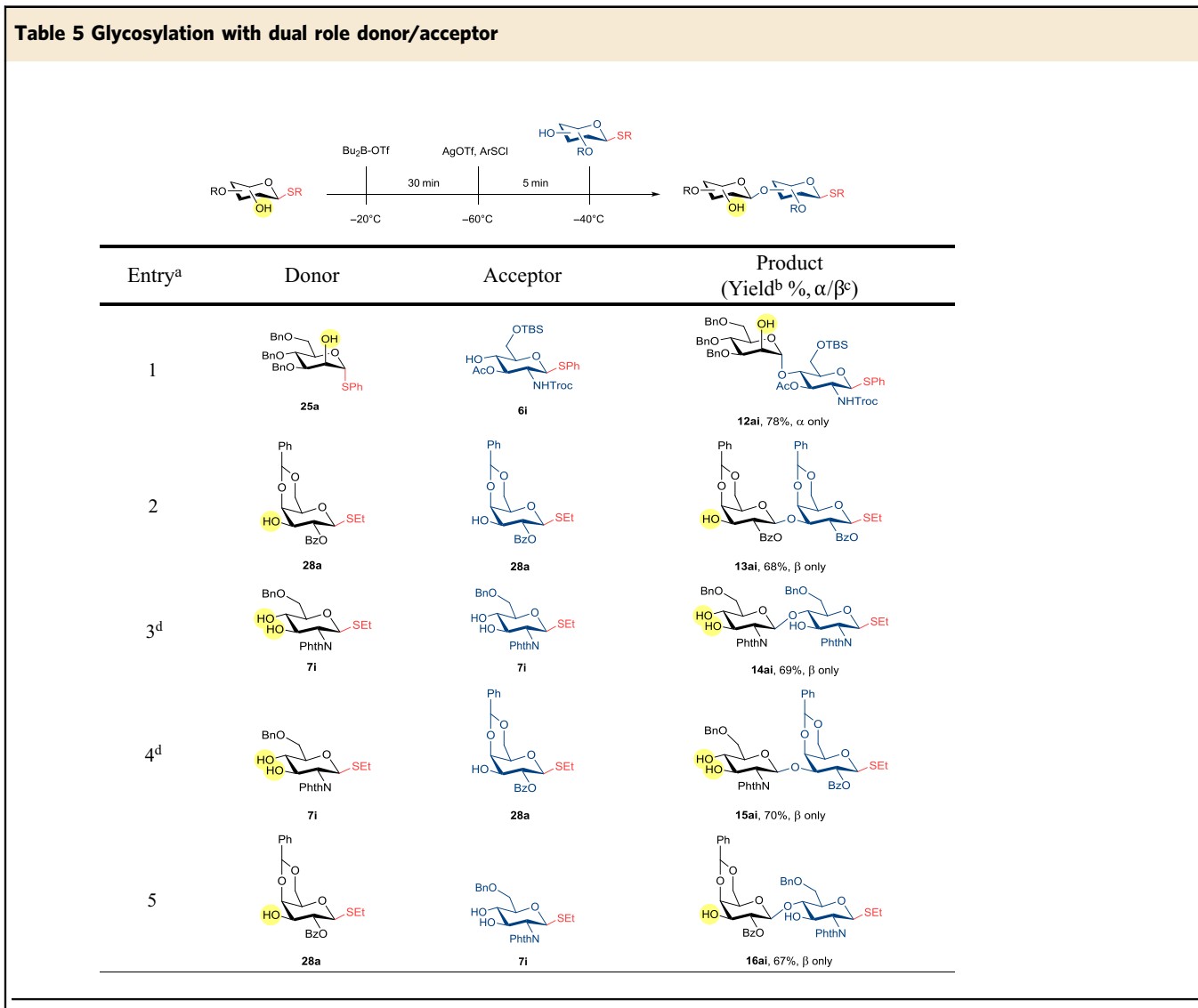

| Entry[a] | Donor | Acceptor | Product (Yield[b] %, α/β[c]) |
|---|---|---|---|
| 1 | 25a | 6i | 12ai, 78%, α only |
| 2 | 28a | 28a | 13ai, 68%, β only |
| 3[d] | 7i | 7i | 14ai, 69%, β only |
| 4[d] | 7i | 28a | 15ai, 70%, β only |
| 5 | 28a | 7i | 16ai, 67%, β only |

[a]Unless otherwise specified, all reactions were carried out with 1 equivalent of donor, 3 eq. of AgOTf, 3 eq. of TTBP, 1.1 eq. of Bu₂BOTf, 1.2 eq. of p-NO₂PhSCl, 1.2 eq. of glycosyl acceptor in 2 mL of solvent for 12 h
[b]isolated yield
[c]determined by ¹H-NMR integration

anomer (Table 3, entry 8). The complex interactions between neighboring OHs were more pronounced with glucosyl donor carrying three free hydroxyls (Table 3, entries 9–12) and the selectivity was generally lower, except for 2,4,6-OHs glucosyl donor **22a** (Table 3, entry 10). Unexpectedly, protecting of the primary 6-OH in glucosyl donor **24a** led to poor selectivity in product **24e** (Table 3, entry 12). Still, in all cases the masking effect of boron was complete and only desired product was detected. Trials with glucosyl donor carrying four hydroxyl groups was unfruitful due to low solubility.

The encouraging results prompted us to prepare substrates for disaccharide surveys. To our delight, many commonly encountered glycosidic bonds proceeded smoothly to provide disaccharides in high yield and selectivity (Table 4, entries 1–5). Epimeric forms of glucose were prepared and the reaction under optimized conditions were compared. As expected, both C-2 epimers glucosyl **1a** and mannosyl **25a** delivered the 1,2-*trans* product **6ai** and **7ai**, respectively (Table 4, entries 6–7). With C-3 epimer products glucoside **8ai** and alloside **9ai**, only β-anomers were detected (Table 4, entries 8–9). For comparison, 2,3,4,6-tetra-*O*-benzyl glucosyl donor **1h** would react to form disaccharide with

high β-selectivity (1α/5β) whereas 2,3,4,6-tetra-*O*-benzyl allosyl donor gave a low selectivity of 1α/1β ratio, consistent with previous reports[52]. The equatorial 3-*O*-Boron complex in the case of glucosyl donor **4a** was inferred to cause minimal steric crowding in β-face of anomeric C-1 and the enhancement of β-preference was likely from a combination of other factors. On the other hand, the axially positioned 3-*O*-Boron complex generated prior to activation of allosyl donor **26a** should provide a substantial shielding effect on the α-face of anomeric C-1, resulting in higher β-selectivity. Initial trials with C-4 epimers glucosyl **7a** and galactosyl **27a** returned similar stereoselective results to their perbenzylated versions, indicated the masking effect was successful without appreciable remote directing effect (Table 4, entries 10–11). To our delight, usage of bulkier dicyclohexyl boron triflate resulted in β-**10ai** as the sole product. Unfortunately, this reagent was unable to improve the selectivity of **11ai** (1α/2.5β). Our NMR studies suggested the boron-masked groups exhibited an electron-withdrawing effect similar to acetyl or benzoyl ester. Kim's group recently published their comprehensive findings on the directing effect by remote electron-withdrawing protecting groups at O-3, O-4 and O-6 positions of

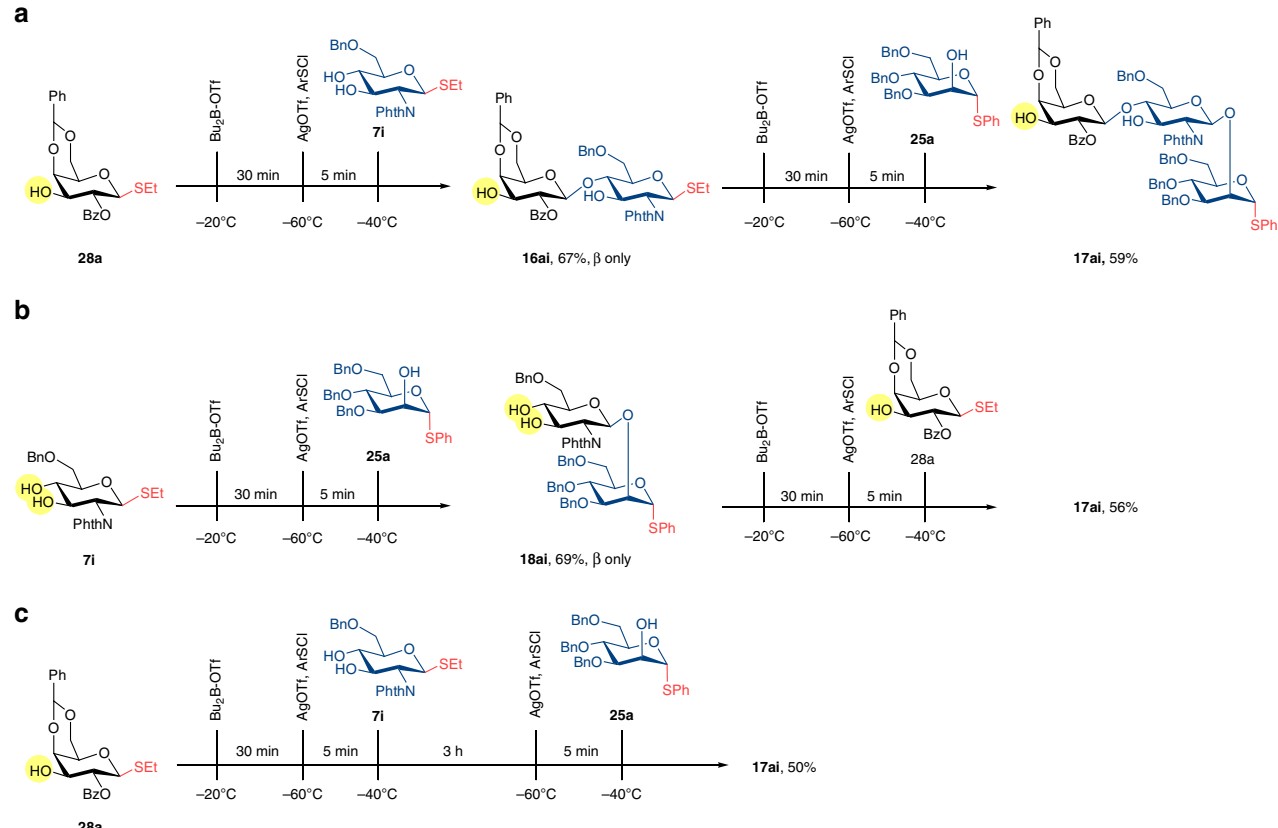

**Fig. 4** Three possible routes to synthesize trisaccharide 17ai. **a**, first donor was galactosyl **28a**. **b**, first donor was glucosamine **7i**. **c**, one-pot synthesis with first donor **28a**

donors in glucosylations[53], galactosylations[53] & mannosylations[54]. Our stereochemical outcomes closely followed the trend observed by Kim's group, in particularly the β-directing effect at O-3 and O-4 positions of glucosyl donors. With our current results, we believed the stereoselectivities are largely influenced by the electron-withdrawing effect of boron complex and its spatial occupancy.

We set out to incorporate this chemistry into a streamlined protocol for design and preparation of important structural scaffolds, which were essential for more complex glycoconjugate synthesis. Our early results were demonstrated in Table 5. Activation of boron-masked thiomannoside **25a** with stoichiometric amount of *p*-nitrobenzenesulfenyl chloride and excess silver trifluoromethanesulfonate, followed by introduction of thioglucoside **6i** delivered product **12ai** (Table 5, entry 1). This compound preserved both thiophenyl on glucose as well as free 2-OH on mannose intact. Since it was possible to use glycosyl donor with unrestricted hydroxyl groups, the same compound could be used as glycosyl acceptor. This presented new opportunities to build diverse library of oligosaccharides in a direct "one step-one glycosylation", starting from the same building blocks. Herein exemplified, compound **28a** and **7i** could be interchangeably assigned the role of glycosyl donor or acceptor, or both, to obtain all four disaccharides (Table 5, entries 2–5). Notably, this flexibility was possible even with employment of a single type of thioglycoside as well as generic protective groups at only non-strategic location. To further expand on this concept, synthesis of trisaccharide **17ai** was achieved via three possible routes (Fig. 4). Firstly, glucosamine **7i** was used as the acceptor to couple with galactosyl donor **28a** to provide disaccharide **16ai**, which was immediately used as the donor to react with mannosyl acceptor

**25a** to furnish trisaccharide **17ai** (Fig. 4a). Alternatively, glucosamine **7i** was used as the donor to couple with mannosyl acceptor **25a** to provide disaccharide **18ai**, which was immediately used as the acceptor to react with galactosyl donor **28a**, furnished the identical trisaccharide **17ai** (Fig. 4b). Finally, trisaccharide **17ai** was successfully prepared in an one-pot glycosylation protocol. Following complete consumption of donor **28a**, a second equivalent of *p*-NO₂PhSCl and freshly dried AgOTf were delivered to reaction mixture, and mannosyl acceptor **25a** was added thereafter. Compound **17ai** was isolated in 50% overall yield (Fig. 4c). No hydrolysis of acceptors or products as well as any self-condensation such as **13ai** and **14ai** were found. The moderate yield of **17ai** was thought to stem from the less reactive acceptors since the only side product was hydrolysis of donor and employment of molecular sieves only improved the overall yield marginally.

In summary, we report an attractive and practical framework to construct glycosidic bonds with high stereoselectivity and regioselectivity without the need for orthogonal protection/deprotection or specialized leaving groups. While confident control of stereoselectivity has been galvanizing research into the field for generations, streamline of working strategies are increasingly important as the complexity quickly rises with every additional sugar being added into the growing glycan chain. We believe the liberation of protection/deprotection steps through temporal boron-masked chemistry would be of great value to more sophisticated strategies such as reactivity-based glycosylation or orthogonal one-pot activation, which relied heavily on a strict and judicious choice of protective groups to achieve the desired selectivity. Further mechanistic studies as well as refining

the method for preparation of various glycoconjugates are ongoing in our laboratory.

## Methods

**General**. The synthesis and characterization of new compounds are provided in the Supplementary Methods. For $^1$H-NMR and $^{13}$C-NMR spectra of the compounds in this article, see Supplementary Figs 1–92. Glycosidic linkage positions of **14ai**, **16ai** and **17ai** were determined based on analysis of $^1$H, $^1$H-$^1$H COSY, 1D-TOCSY, $^{13}$C-$^1$H HSQC. Anomeric configuration of obtained mannosides **4ai**, **7ai**, **12ai** were determined based on chemical shifts of anomeric protons and coupling constants of $^{13}$C1-$^1$H1. For preliminary results, see Supplementary Discussions.

**Experimental procedure**. All reactions were conducted under an atmosphere of nitrogen, unless otherwise indicated. Anhydrous solvents were transferred via oven-dried syringe. Flasks were flame-dried and cooled under a stream of nitrogen. All reagents and solvents were obtained from commercial suppliers and used without further purification unless otherwise stated. Chromatograms were visualized by fluorescence quenching with UV light at 254 nm or by staining using a basic solution of potassium permanganate. Evaporation of organic solutions was achieved by rotary evaporation with a water bath temperature below 40 °C. Product purification by flash column chromatography was accomplished using silica gel 60 (0.010–0.063 mm). Technical grade solvents were used for chromatography and distilled prior to use. Optical rotations were measured in CHCl$_3$ with a 1 cm cell (c given in g 100 mL$^{-1}$). Melting points were obtained in open capillary tubes in melting point apparatus. IR spectra were recorded using FTIR and reported in cm$^{-1}$. High resolution mass spectra (HRMS) were recorded on Q-TOF mass spectrometer. Accurate masses are reported for the molecular ion [M + H]$^+$ or a suitable fragment ion. NMR spectra were recorded at room temperature on a 400 MHz and 500 MHz NMR spectrometer. The residual solvent signals were taken as the reference (726 ppm for $^1$H-NMR spectroscopy and 77.23 ppm for $^{13}$C-NMR spectroscopy). Chemical shifts are reported in delta ($\delta$) units, parts per million (ppm) downfield from trimethylsilane (TMS). Chemical shift ($\delta$) is referred in terms of ppm, coupling constants (J) are given in Hz. Following abbreviations classify the multiplicity: s = singlet, d = doublet, dd = doublet of doublet, t = triplet, q = quartet, m = multiplet, br = broad or unresolved.

**Data availability**. The authors declare that the data supporting the findings of this study are available within the article and its Supplementary Information files. All data are available from the authors upon reasonable request.

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

## Acknowledgements

Financial support from the Ministry of Education (MOE 2013-T3-1-002), National Research Foundation (NRF2016NRF-NSFC002-005) and Nanyang Technological University (RG14/16), Singapore are gratefully acknowledged.

## Author contributions

L.M. H.K. and L.X.-W. designed the project and wrote the manuscript. L.M.H.K. carried out the synthetic work. H.J.X. and B.G. assisted in preparation of several glycosyl donors for the study. All authors discussed the results and commented on the manuscript.

## Additional information

**Competing interests:** The authors declare no competing financial interests.

