## [Peer Review File · Nature Communications]

Reviewers' comments:

Reviewer #1 (Remarks to the Author):

The manuscript by Liu and co-workers reports on the authors studies into boron mediated glycosylation reactions utilizing partially protected glycosyl donor and acceptor substrates. The authors report good yields for the glycosylated products with reasonable control over the stereochemistry and regioselectivity of the newly formed glycosidic bond. One of the primary motivations for carrying out this research is to limit the number of protecting group manipulations required for oligosaccharide synthesis, however the partially protected systems reported still require significant protecting group manipulation. Although the authors have certainly demonstrated an efficient methodology for the synthesis of disaccharides (and a single example of a trisaccharide) and have applied it to a useful range of substrates, a significant number of partially protected glycosyl acceptor and donor strategies have previously been reported by other groups. Although these results will be of interest in glycoscience the authors have not made a compelling case that the report is of sufficient novelty to justify publication in Nature Communications. The results would be suitable for publication in an alternative journal.

Reviewer #2 (Remarks to the Author):

This manuscript by Liu et al. deals with stereoselective glycosylation with partial protected thioglycosyl donors utilizing dialkylboryl triflates as *in situ* masking reagent. In this study, the authors found that hydroxyl group masked by dialkylboryl triflate did not act as glycosyl acceptor. In addition, the masking group could tolerate under the glycosylation reaction conditions using thiophilic activator (AgOTf/*p*-NO₂PhSCl, Ph₂SO/Tf₂O, DMTSF, etc) without causing detrimental effects on reaction yield and stereoselectivity. Based on these findings, they established a relatively simple and efficient glycosylation protocol. So basically, incubation of thioglycosyl donor carrying one to three unprotected hydroxyl groups and a slightly excess amount (to the free hydroxyl groups) of dialkylboryl triflate, followed addition of thiophilic activator and glycosyl acceptor afforded the corresponding glycosides with high regio- and generally good stereoselectivity. Furthermore, they revealed this method has wide substrate scope. I think this glycosylation method is interesting and attractive not only in carbohydrate chemistry but also in organic chemistry. However, unfortunately, the proposed reaction mechanism is highly speculative and not fully supported. In addition, some experimental results were not theoretically explained. Therefore, I recommend publication of this manuscript in *Nature Communications* only after addressing concerns mentioned listed below:

1. Spectral evidences that free OH groups on glycosyl donors were completely masked by dialkylboryl triflates should be shown in the supporting information.
2. Solvent effect on this glycosylation is unclear (Table 1, entries 4-8). Although EtCN is compatible in the reaction, MeCN is not. What is happened in the case of MeCN ?
3. Although the authors proposed that the boron complex is acting like a directing group through its spatial occupancy (Figure 2), no appreciable change was found in selectivity when other dialkyl boron triflates were used (page 6, line 8). Similarly, in the case of **11a** (Table 4, entry 11), bulky dicyclohexyl boron triflate did not affect on the stereoselectivity. Therefore, at this stage, I can't believe their hypothesis.

4. When 3-OH glucosyl donors were used, complete beta stereoselectivities were observed (Table 2, entries 4 and 5). What is the rationale for increasing stereoselectivity compared to the control experiment using **1h** (Figure 3a).

5. Although the authors mentioned that "It is probable the acyl group formed a complex with boron center, thus attenuated the directing effect of dialkylboron" (page 7, line 7), spectral evidences (such as ¹H and ¹¹B NMR spectra) that acyl group formed a complex should be shown in the supporting information.

6. Although the authors mentioned that "In a control experiment, this sole product was only detected in trace....."(page 7, line 13), what is happened in the control experiment ? DMTSF did not activate donor **9a** ?

7. Although glucosyl donor **12a** was converted into 1,6-anhydroglucoside **12e** (Table 2, entry 12), I could not understand whether 6-OH was not masked by Bu₂BOTf or alcohol exchange reaction with BuOH, followed intramolecular reaction took place rapidly.

8. At page 10, line 3, a synergistic influence is unclear. The authors should explain more clearly how the masking groups affect on the stereoselectivity. This point is very important to predict the stereoselectivity for the glycosylation reactions.

9. The chemical yields of **17ai** (Table 5, entries 6 and 8) were moderate (59% and 56% yields, respectively). Please explain the reason in the manuscript. Did any undesired side reactions occur ? In addition, the authors should mention how they determined the glycosidic linkage position of **14ai**, **16ai**, and **17ai** in the manuscript.

Minor points

1. Although the authors mentioned that "Using dialkylboron triflate as *in situ*.....with complete control of regioselectivity and stereoselectivity" at the abstract, stereoselectivity is not complete.

2. At page 6, Table 1, entry 8, alpha:beta =1:5 is incorrect.

3. At page 10, line 8, "masking of the primary 6-OH" should be changed to "protecting of the primary 6-OH".

4. At page 13, Table 5, entry 1, "beta only" is probably incorrect. In addition, the authors should mention how they determined the anomeric configuration of the obtained mannosides **4ai**, **7ai**, and **12ai** in the manuscript.

Reviewer #3 (Remarks to the Author):

The manuscript by Liu and co-workers describes a stereo and regioselective glycosylation strategy with unprotected glycosyl donors. Protection and deprotection steps make oligosaccharide synthesis a tedious work, one solution to this problem is to use protection-free strategy. However, huge challenges exist in application of this strategy in oligosaccharide synthesis, especially when employing unprotected

glycosyl donors in glycosylation reactions. The present manuscript smartly introduced boron reagents as temporary groups to protect the free hydroxy groups of glycosyl donors, which allowed the glycosylation reaction proceeded smoothly without touching these hydroxy groups. These temporary groups were easily removed during the work-up process to release the hydroxy groups which could participate further glycosylation reaction to elongation the carbohydrate chains. Wide range of substrates were examined and high efficiency were observed even for those donors with multi free hydroxy groups. Most importantly, the temporal group located in C-2 position also played as a masking group which allowed the challenging 1,2-trans glycosylation (especially beta glycosylation). Overall, the authors developed an elegant protection-free glycosylation strategy, the novelty and synthetic potential of this strategy should be of considerable interest to the synthetic carbohydrate community as well as more general scientific field. Thus, I highly recommend its publication in the Nature Communication after the following issues have been addressed.

1. The authors suggested that the 1,2-trans selectivity was resulted from the masking effect of the temporary protecting groups. Is it possible that a 1,2-anhydro sugar intermediate formed during the activation process? It is this intermediate promoted the 1,2-trans selectivity? Especially, if looked at table 2, entry 12, a high yielding of 1,6-anhydro sugar was observed, this possibility could not be excluded.
2. Some donors without C-2 hydroxy groups, for example, 4a, 7a, 17a and 19a, still gave good beta selectivity, what is the driving force for this unusual selectivity? apparently, the masking effect or participation effect was not exist for these substrates.
3. How's the strength of the boron reagent linking to the hydroxy group? Is it possible to break the O-B bond to release the hydroxy group once the glycosylation reaction finished, then the released hydroxy group could participate next glycosylation reaction in the same pot?
4. A lot of examples were presented by utilizing butanol as acceptor, however, butyl glycosides are not real oligosaccharides, most of these examples could be moved to supporting information to make the manuscript pithier. In addition, it is confusing that a 1:5 of alfa to beta ratio was recorded while the yield was n.d. in table 1, entry 8. The description of the three synthetic routes to trisaccharide 17ai was unclear, it is better to present these result in a single scheme. By the way, the presentation of alfa to beta ratio in tables ("x alfa: y beta") was not commonly used way.

Reviewers' comments:

Reviewer #1 (Remarks to the Author):

The manuscript by Liu and co-workers reports on the authors studies into boron mediated glycosylation reactions utilizing partially protected glycosyl donor and acceptor substrates. The authors report good yields for the glycosylated products with reasonable control over the stereochemistry and regioselectivity of the newly formed glycosidic bond. One of the primary motivations for carrying out this research is to limit the number of protecting group manipulations required for oligosaccharide synthesis, however the partially protected systems reported still require significant protecting group manipulation. Although the authors have certainly demonstrated an efficient methodology for the synthesis of disaccharides (and a single example of a trisaccharide) and have applied it to a useful range of substrates, a significant number of partially protected glycosyl acceptor and donor strategies have previously been reported by other groups. Although these results will be of interest in glycoscience the authors have not made a compelling case that the report is of sufficient novelty to justify publication in Nature Communications. The results would be suitable for publication in an alternative journal.

→We are thankful for the reviewer's impression of our original manuscript. We have conducted more experiments to gain concrete information about the reaction mechanism and have try our best to improve the manuscript. We hope that the reviewer will share the same sentiments we received from other reviewers that this work is of great interest to glycoscience as well as general scientific field.

Reviewer #2 (Remarks to the Author):

This manuscript by Liu et al. deals with stereoselective glycosylation with partial protected thioglycosyl donors utilizing dialkylboryl trifrates as in situ masking reagent. In this study, the authors found that hydroxyl group masked by dialkylboryl trifrate did not act as glycosyl acceptor. In addition, the masking group could tolerate under the glycosylation reaction conditions using thiophilic activator (AgOTf/p-NO₂PhSCl, Ph₂SO/Tf₂O, DMTSF, etc) without causing detrimental effects on reaction yield and stereoselectivity. Based on these findings, they established a relatively simple and efficient glycosylation protocol. So basically, incubation of thioglycosyl donor carrying one to three unprotected hydroxyl groups and a slightly excess amount (to the free hydroxyl groups) of dialkylboryl trifrate, followed addition of thiophilic activator and glycosyl acceptor afforded the corresponding glycosides with high regio- and generally good stereoselectivity. Furthermore, they revealed this method has wide substrate scope. I think this glycosylation method is interesting and attractive not only in carbohydrate chemistry but also in organic chemistry. However, unfortunately, the proposed reaction

mechanism is highly speculative and not fully supported. In addition, some experimental results were not theoretically explained. Therefore, I recommend publication of this manuscript in Nature Communications only after addressing concerns mentioned listed below:

→ We are very grateful for the reviewer's detailed discussion and insightful suggestions, which help us to obtain more concrete information about the reaction mechanism and improve our manuscript.

1. Spectral evidences that free OH groups on glycosyl donors were completely masked by dialkylboryl triflates should be shown in the supporting information.

→ We conducted low temperature NMR for the following donors:

Upon addition of Bu_2BOTf at $-40\text{ }^\circ\text{C}$, the proton of OH at C-2 disappeared and H-2 was shifted downfield to 4.20 ppm from its previous position at 3.56 ppm ($\Delta=0.64\text{ ppm}$), whereas other protons remained relatively unchanged. The displacement of OH proton with boron complex exerted a deshielding effect on its immediate vicinal proton. This suggested the boron complex having an electron-withdrawing effect similar to acetyl, benzoyl, etc... This intermediate was found to be stable after all NMR experiments were collected (1D-TOCY, COSY, ^{11}B) and only showed decomposition at or above $-10\text{ }^\circ\text{C}$. Interestingly, ^{11}B -NMR showed a broad signal with peak at about -3.0 ppm . This value was typical of a tetravalent boron species. In a second run, only 0.5 equivalent of Bu_2BOTf was added and we observed a mixture of boron-masked and original **2.2a**. **This excluded the formation of a dimer of the type RO-B(Bu₂)-OR, which led us to propose the observed intermediate to be RO-B(Bu₂)-OTf. It is noteworthy that there is usually excess amount of triflate ion in our reaction condition (e.g. from Bu_2BOTf , Tf_2O , AgOTf ...).** In a third run, one equivalent of BuOH was added 5 mins after introduction of one equivalent of Bu_2BOTf . We observed no change in chemical shift of boron-masked **2.2a** after 30 mins at $-40\text{ }^\circ\text{C}$. However, excess amount of BuOH resulted in mixture of boron-masked and original **2.2a**. We concluded that the exchange process between different alcohol species was slow enough at low temperature to warrant the desired masking effect. See Supplementary Figure 94-96.

^1H and ^{11}B NMR of donor **2.2b** was obtained following the protocol for **2.2a**. We observed the proton of OH at C-2 disappeared and H-2 was shifted downfield to 4.30 ppm from its

previous position at 3.56 ppm ($\Delta=0.74$ ppm). In addition, H-3 was shifted appreciably downfield to 5.48 ppm from its previous position at 5.26 ppm ($\Delta=0.22$ ppm). ^{11}B -NMR showed a broad signal with peak at about -20 ppm. The change in chemical shifts of H-2, H-3 and ^{11}B can thus be attributed to the electron-withdrawing effect of boron complex and possible attenuating effect of neighboring acetyl group. See Supplementary Figure 97-99.

2.2c

^1H and ^{11}B NMR of donor **2.2c** was obtained following the protocol for **2.2a**. We observed the proton of OH at C-3 disappeared and H-3 was shifted downfield to 4.50 ppm from its previous position at 3.83 ppm ($\Delta=0.67$ ppm). In addition, H-2 was shifted appreciably downfield to 5.10 ppm from its previous position at 4.94 ppm ($\Delta=0.16$ ppm). ^{11}B -NMR showed a broad signal with peak at about -21 ppm. The change in chemical shifts of H-2, H-3 and ^{11}B can thus be attributed to the electron-withdrawing effect of boron complex and possible attenuating effect of neighboring acetyl group. See Supplementary Figure 100-102.

2. Solvent effect on this glycosylation is unclear (Table 1, entries 4-8). Although EtCN is compatible in the reaction, MeCN is not. What is happened in the case of MeCN ?

→ This is probably a misunderstanding. We used Nitroethane ($\text{CH}_3\text{CH}_2\text{NO}_2$, Table 1, entry 6) and Acetonitrile (CH_3CN , Table 1, entry 8). We did not use EtCN. For acetonitrile, we only obtained trace amount of product **1e**. The donor was recovered. We believe the reaction was too slow with this solvent/activator combination.

3. Although the authors proposed that the boron complex is acting like a directing group through its spatial occupancy (Figure 2), no appreciable change was found in selectivity when other dialkyl boron triflates were used (page 6, line 8). Similarly, in the case of **11ai** (Table 4, entry 11), bulky dicyclohexyl boron triflate did not affect on the stereoselectivity. Therefore, at this stage, I can't believe their hypothesis.

→ We believe the steric size of the acceptor also influenced the stereo-outcome of the reaction as the difference between dialkylboron triflates may not be enough to secure the stereoselectivity for small alcohols such as methanol. For this study, we are focusing on demonstrating the use of temporal masking of unprotected hydroxyl groups at remote positions, which are compatible with another directing group, e.g. acetyl, benzoyl, at C-2 position (e.g. product **8e**, **11e**, **20e**, **13ai-17ai**). In the case of donors having unprotected OH at C-2, we discovered that the dialkylboron triflate can provide 1,2-trans product directly.

4. When 3-OH glucosyl donors were used, complete beta stereoselectivities were observed (Table 2, entries 4 and 5). What is the rational for increasing stereoselectivity compared to the control experiment using 1h (Figure 3a).

→ NMR studies suggested the boron-masked groups exhibited an electron-withdrawing effects similar to acetyl or benzoyl ester. Kim's group recently published their comprehensive findings on the directing effect by remote electron-withdrawing protecting groups at O-3, O-4 and O-6 positions of donors in glucosylations, galactosylations and mannosylations (Ref 50: *Tetrahedron*. **71**, 5315-5320 (2015) and Ref 51: *J. Am. Chem. Soc.* **131**, 17705-17713 (2009)). Our stereochemical outcomes closely followed the trend observed by Kim's group, in particularly the β -directing effect at O-3 and O-4 positions of glucosyl donors. With our current results, we believe the stereoselectivities are largely influenced by the electron-withdrawing effect of boron complex and its spatial occupancy. Dedicated study to fully map out the complex interactions during a glycosylation reaction with boron-masked reagent is in progress.

5. Although the authors mentioned that "It is probable the acyl group formed a complex with boron center, thus attenuated the directing effect of dialkylboron" (page 7, line 7), spectral evidences (such as ^1H and ^{11}B NMR spectra) that acyl group formed a complex should be shown in the supporting information.

→ We have obtained ^1H , ^{11}B NMR spectra for donor with 2-OH-3-O-acetyl and 3-OH-2-O-acetyl. ^{11}B -NMR experiments revealed an upfield shift to about -20 ppm compared to the chemical shift of -3 ppm with donor **1a**, suggesting the formation of a different tetravalent boron intermediate. We proposed that the acyl group formed a complex with boron center, thus attenuated the steric effect of dialkylboron.

6. Although the authors mentioned that "In a control experiment, this sole product was only detected in trace....."(page 7, line 13), what is happened in the control experiment ? DMTSF did not activate donor 9a ?

→ This is probably a misunderstanding. We have rewritten the discussion: "If no boron was added and DMTSF was used as activating agent, a complicated mixture was observed and only trace amount of **9e** was isolated (<5%)."

7. Although glucosyl donor 12a was converted into 1,6-anhydroglucoside 12e (Table 2, entry 12), I could not understand whether 6-OH was not masked by Bu₂BOTf or alcohol exchange reaction with BuOH, followed intramolecular reaction took place rapidly.

→ In one of the NMR studies, one equivalent of BuOH was added 5 mins after introduction of one equivalent of Bu₂BOTf. We observed no change in chemical shift of boron-masked **2.2a**

after 30 mins at $-40\text{ }^{\circ}\text{C}$. We concluded that the exchange process between different alcohol species was slow enough at low temperature to warrant the desired masking effect.

8. At page 10, line 3, a synergistic influence is unclear. The authors should explain more clearly how the masking groups affect on the stereoselectivity. This point is very important to predict the stereoselectivity for the glycosylation reactions.

→ Our NMR studies suggested the boron-masked groups exhibited an electron-withdrawing effects similar to acetyl or benzoyl ester. Our stereochemical outcomes closely followed the trend observed by Kim's group, in particularly the β -directing effect at O-3 and O-4 positions of glucosyl donors having electron-withdrawing groups. We have moved the discussion to the end of Table 3.

9. The chemical yields of 17ai (Table 5, entries 6 and 8) were moderate (59% and 56% yields, respectively). Please explain the reason in the manuscript. Did any undesired side reactions occur? In addition, the authors should mention how they determined the glycosidic linkage position of 14ai, 16ai, and 17ai in the manuscript.

→ No hydrolysis of acceptors or products as well as any self-condensation such as **13ai** and **14ai** were found. The moderate yield of **17ai** was thought to stem from the less reactive acceptors since the only side product was hydrolysis of donor and employment of molecular sieves only improved the overall yield marginally. Glycosidic linkage positions of **14ai**, **16ai** and **17ai** were determined based on analysis of ^1H , ^1H - ^1H COSY, 1D-TOCSY, ^{13}C - ^1H HSQC. We added this in the Methods section of the manuscript.

Herein, we present an example with compound **14ai**

Based on ^{13}C - ^1H HSQC, we can assign the anomeric proton and carbon of *O*-linked glucosamine and *S*-linked glucosamine. 1D-TOCSY selective irradiation on anomeric proton of each glucosamine unit revealed the chemical shifts of its isolated spin system. Overlaying this information on ^1H - ^1H COSY and we are able to determine the glycosidic linkage position as GlcNAc- β -(1 \rightarrow 4)-GlcNAc- β -SEt.

^{13}C - ^1H HSQC of compound 14ai

^1H - ^1H COSY with 1D-TOCSY overlay of individual glucosamine unit.

Minor points

1. Although the authors mentioned that “Using dialkylboryl triflate as in situ.....with complete control of regioselectivity and stereoselectivity” at the abstract, stereoselectivity is not complete.

→ It was changed to “with good control of regioselectivity and stereoselectivity”

2. At page 6, Table 1, entry 8, alpha:beta =1:5 is incorrect.

→ We apologize for the typo. We obtained trace amount of product **1e**. The donor was recovered. We believe the reaction was too slow with this solvent/activator combination.

3. At page 10, lane 8, “masking of the primary 6-OH” should be changed to “protecting of the primary 6-OH”.

→ The change was added.

4. At page 13, Table 5, entry 1, “beta only” is probably incorrect. In addition, the authors should mention how they determined the anomeric configuration of the obtained mannosides **4ai**, **7ai**, and **12ai** in the manuscript.

→ We apology for the typo. Anomeric configuration of obtained mannosides **4ai**, **7ai**, **12ai** were determined based on chemical shifts of anomeric protons and coupling constants of $^{13}\text{C1-}^1\text{H1}$. Usually, β anomer has $^1J_{\text{C1,H1}} \sim 160\text{Hz}$ whereas α anomer has $^1J_{\text{C1,H1}} \sim 170\text{Hz}$. We added this in the Methods section of the manuscript.

Herein, we present an example with compound **12ai** (Bruker ^{13}C , 125MHz). Calculation of the coupling constant for mannose $^1J_{\text{C1-H1}} = 170.83\text{ Hz}$ and glucosamine $^1J_{\text{C1-H1}} = 161.25\text{ Hz}$. Hence mannose has alpha configuration and glucosamine has beta configuration.

Reviewer #3 (Remarks to the Author):

The manuscript by Liu and co-workers describes a stereo and regioselective glycosylation strategy with unprotected glycosyl donors. Protection and deprotection steps make oligosaccharide synthesis a tedious work, one solution to this problem is to use protection-free strategy. However, huge challenges exist in application of this strategy in oligosaccharide synthesis, especially when employing unprotected glycosyl donors in glycosylation reactions. The present manuscript smartly introduced boron reagents as temporary groups to protect the free hydroxy groups of glycosyl donors, which allowed the glycosylation reaction to proceed smoothly without touching these hydroxy groups. These temporary groups were easily removed during the work-up process to release the hydroxy groups which could participate in further glycosylation reactions to elongate the carbohydrate chains. A wide range of substrates were examined and high efficiency was observed even for those donors with multiple free hydroxy groups. Most importantly, the temporary group located in the C-2 position also played as a masking group which allowed the challenging 1,2-trans glycosylation (especially beta glycosylation). Overall, the authors developed an elegant protection-free glycosylation strategy, the novelty and synthetic potential of this strategy should be of considerable interest to the synthetic carbohydrate

community as well as more general scientific field. Thus, I highly recommend its publication in the Nature Communication after the following issues have been addressed.

→we would like to thank the reviewer for his recommendation and the insightful suggestions, which help us to improve the manuscript.

1. The authors suggested that the 1,2-trans selectivity was resulted from the masking effect of the temporary protecting groups. Is it possible that a 1,2-anhydro sugar intermediate formed during the activation process? It is this intermediate promoted the 1,2-trans selectivity? Especially, if looked at table 2, entry 12, a high yielding of 1,6-anhydro sugar was observed, this possibility could not be excluded.

→We prepared per-benzylated 1,2-anhydro glucose following standard protocol (Cheshev *et al. Carbohydr. Res.* 341, 2714-2716 (2006)). The compound was subjected to various conditions:

Premixing donor **2.2d** before introduction of butanol acceptor delivered the compound **1e** in 65% yield but with no stereoselectivity preference $\alpha/\beta = 1.1/1$. On the other hand, premixing butanol and dibutylboron triflate to generate the masked acceptor did not provide **1e**. The only product we obtained was the hydrolyzed compound **2.2e**. The boron reagent is concluded to be able to act as Lewis acid to activate the epoxide **2.2d** but without β -directing effect as observed with donor **1a**. In addition, masking of acceptor with boron prevented its role as nucleophile and the hydrolytic product was generated during workup phase. We believe the reaction is unlikely to generate a 1,2-anhydro intermediate. The 1,6-anhydro sugar was only obtained when the conformation of the donor was 1C_4 which position OH-6 in close proximity to anomeric center. For other OH-6 donor such as **10a** or **11a**, we didn't observe the 1,6-anhydro product.

2. Some donors without C-2 hydroxy groups, for example, 4a, 7a, 17a and 19a, still gave good beta selectivity, what is the driving force for this unusual selectivity? apparently, the masking effect or participation effect was not exist for these substrates.

→ We conducted low temperature NMR studies to observe the displacement of hydroxyl protons upon addition of boron reagent. We detected the boron complex exerted a deshielding effect on its immediate vicinal proton. This suggested the boron complex having an electron-withdrawing effect similar to acetyl, benzoyl, etc... Kim's group recently published their comprehensive findings on the directing effect by remote electron-withdrawing protecting groups at O-3, O-4 and O-6 positions of donors in glucosylations, galactosylations and mannosylations. (Ref 50: *Tetrahedron*. **71**, 5315-5320 (2015) and Ref 51: *J. Am. Chem. Soc.* **131**, 17705-17713 (2009)). Our stereochemical outcomes closely followed the trend observed by Kim's group, in particularly the β -directing effect at O-3 and O-4 positions of glucosyl donors. With our current results, we believe the stereoselectivities are largely influenced by the electron-withdrawing effect of boron complex and its spatial occupancy. Dedicated study to fully map out the complex interactions during a glycosylation reaction with boron-masked reagent is in progress.

3. How's the strength of the boron reagent linking to the hydroxy group? Is it possible to break the O-B bond to release the hydroxy group once the glycosylation reaction finished, then the released hydroxy group could participate next glycosylation reaction in the same pot?

→ Low temperature NMR revealed the boron-masked intermediate was stable after all NMR experiments were collected at -40 °C and only showed decomposition at or above -10 °C. We also introduced one equivalent of butanol into the NMR tube and no exchange process was detected after 30mins. However, excess amount of butanol resulted in a mixture of masked- and unmaked-donor. Under our current optimized condition, we believe a workup step to fully break down the O-B bond after a glycosylation reaction is required to liberate the masked hydroxyl group. The masked hydroxyl group however can be preserved through additional glycosylation steps, as demonstrated in our one-pot synthesis of trisaccharide **17ai**.

4. A lot of examples were presented by utilizing butanol as acceptor, however, butyl glycosides are not real oligosaccharides, most of these examples could be moved to supporting information to make the manuscript pithier. In addition, it is confusing that a 1:5 of alfa to beta ratio was recorded while the yield was n.d. in table 1, entry 8. The description of the three synthetic routes to trisaccharide 17ai was unclear, it is better to present these result in a single scheme. By the way, the presentation of alfa to beta ratio in tables ("x alfa: y beta") was not commonly used way.

→ We are content with moving these results into supporting information should the editor found them to go over the space limit. We apologize for the typo in Table 1, entry 8. When acetonitrile was used as the solvent, only trace amount of butyl glucoside **1e** ($\alpha/\beta = 1/5$) was isolated with

most of the donor recovered. We believe the reaction was too slow with this solvent/activator combination. As requested, we present our three synthetic routes to trisaccharide **17ai** in Figure 4 separately. Finally, we change the presentation of alpha to beta ratio, e.g. $\alpha/\beta = 1/5$.

Figure 4 | Three possible routes to synthesize trisaccharide 17ai. a, first donor was galactosyl 28a. b, first donor was glucosamine 7i. c, one-pot synthesis with first donor 28a.

Reviewers' comments:

Reviewer #2 (Remarks to the Author):

Since the authors responded the reviewer's comments (including additional experiments) point by point, I recommend publication of this manuscript after revision of the following points.

1. The authors responded to the reviewer's comment (4. When 3-OH glucosyl donors were used, complete beta stereoselectivities were observed (Table 2, entries 4 and 5). What is the rationale for increasing stereoselectivity compared to the control experiment using 1 h (Figure 3a)) as follows: "NMR studies suggested the boron-masked groups exhibited an electron-withdrawing effects similar to acetyl or benzoyl ester. Kim's group recently published their comprehensive findings on the directing effect by remote electron-withdrawing protecting groups at O-3, O-4 and O-6 positions of donors in glucosylations, galactosylations and mannosylations (Ref 50: Tetrahedron. 71, 5315-5320 (2015) and Ref 51: J. Am. Chem. Soc. 131, 17705-17713 (2009). Our stereochemical outcomes closely followed the trend observed by Kim's group, in particularly the β -directing effect at O-3 and O-4 positions of glucosyl donors. With our current results, we believe the stereoselectivities are largely influenced by the electron-withdrawing effect of boron complex and its spatial occupancy. Dedicated study to fully map out the complex interactions during a glycosylation reaction with boron-masked reagent is in progress."

However, Kim and co-workers reported that production of glycosyl triflate intermediate is important factor for increasing stereoselectivity. Thus, is it possible to detect the corresponding triflate intermediate in low temperature NMR experiments? In addition, is it possible to reveal that the glycosylation proceed in SN2-type manner?

2. The authors responded to the reviewer's comment (7. Although glucosyl donor 12a was converted into 1,6-anhydroglucoside 12e (Table 2, entry 12), I could not understand whether 6-OH was not masked by Bu₂BOTf or alcohol exchange reaction with BuOH, followed intramolecular reaction took place rapidly.) as follows:

"In one of the NMR studies, one equivalent of BuOH was added 5 mins after introduction of one equivalent of Bu₂BOTf. We observed no change in chemical shift of boron-masked 2.2a after 30 mins at -40 °C. We concluded that the exchange process between different alcohol species was slow enough at low temperature to warrant the desired masking effect."

However, since I am asking about the reaction of donor 12a not donor 2.2a, they should answer appropriately.]

3. I recommend the authors to cite relevant papers of the regio- and stereoselective glycosylation using a organoboron reagent as a transient masking group, such as Tetrahedron Lett. 2010, 51, 1570.

Reviewer #3 (Remarks to the Author):

A series of NMR studies and other experiments have been carried out to further demonstrate the reaction mechanisms. The comments provided by the reviewers have been well addressed by the authors. These revisions improved the quality of the paper and now it is suitable for publication in Nature Commun. So, I highly recommend its publication without any further revision.

Reviewers' comments:

Reviewer #2 (Remarks to the Author):

Since the authors responded the reviewer's comments (including additional experiments) point by point, I recommend publication of this manuscript after revision of the following points.

1. The authors responded to the reviewer's comment (4. When 3-OH glucosyl donors were used, complete beta stereoselectivities were observed (Table 2, entries 4 and 5). What is the rationale for increasing stereoselectivity compared to the control experiment using 1 h (Figure 3a)) as follows:

“NMR studies suggested the boron-masked groups exhibited an electron-withdrawing effects similar to acetyl or benzoyl ester. Kim's group recently published their comprehensive findings on the directing effect by remote electron-withdrawing protecting groups at O-3, O-4 and O-6 positions of donors in glucosylations, galactosylations and mannosylations (Ref 50: Tetrahedron. 71, 5315-5320 (2015) and Ref 51: J. Am. Chem. Soc. 131, 17705-17713 (2009)). Our stereochemical outcomes closely followed the trend observed by Kim's group, in particularly the β -directing effect at O-3 and O-4 positions of glucosyl donors. With our current results, we believe the stereoselectivities are largely influenced by the electron-withdrawing effect of boron complex and its spatial occupancy. Dedicated study to fully map out the complex interactions during a glycosylation reaction with boron-masked reagent is in progress.”

However, Kim and co-workers reported that production of glycosyl triflate intermediate is important factor for increasing stereoselectivity. Thus, is it possible to detect the corresponding triflate intermediate in low temperature NMR experiments? In addition, is it possible to reveal that the glycosylation proceed in SN2-type manner?

→ We have conducted the low temperature NMR experiments with donor **4a** having 3-OH. Diphenyl sulfoxide was included to mirror the reaction condition as in Table 2, entry 4.

4a

Upon addition of Bu_2BOTf at $-40\text{ }^\circ\text{C}$, the proton of OH at C-3 disappeared and H-3 was shifted downfield to 4.37 ppm from its previous position at 3.44 ppm ($\Delta=0.93$ ppm), whereas other protons remained relatively unchanged. The comparison between **4a** at $-40\text{ }^\circ\text{C}$ and the boron-masked **4a.1** was shown in Figure 1.

Figure 1. ^1H -NMR of donor **4a** and the boron-masked **4a.1** at $-40\text{ }^\circ\text{C}$.

Temperature was lowered to $-65\text{ }^\circ\text{C}$. Shimming was repeated and ^1H -NMR was collected and the chemical shifts remained stationary. Next, the NMR tube was lifted up a second time to add triflic anhydride (1M in CH_2Cl_2), shaken up and quickly descended back into the NMR probe (we noticed a slightly brown solution already formed). Compound **4a.1** was found to transform quantitatively into a single new compound **4a.2**, characterized by its anomeric proton signal, a broad singlet at δ 6.21. The ^{13}C -NMR also indicated clean formation of a single new carbohydrate with anomeric carbon signal at δ 108.6 (^1H , ^{13}C HSQC correlation). An ^{19}F NMR revealed a number of signals at δ 4.26, 0.12 and -2.76 . Those at δ 4.26 and -2.76 were assigned to tri-*tert*-butyl-pyrimidinium triflate and Tf_2O , respectively, with the aid of control samples. Thus the signal at δ 0.12 was assigned as coming from the triflate intermediate **4a.2**. Coupling constants of ^{13}C - ^1H 1 was calculated at 184Hz. Based on the chemical shifts from various atomic spectra and coupling constants we assigned the new species **4a.2** to be the alpha-triflate intermediate. Our results were compared with similar triflate intermediate as reported in various literatures (Crich, D. et al. *JACS*, **1997**, *119*, 11217–11223; Yoshida et. al. *ACIE*. **2004**, *43*, 2145–2148 and Kim, K. S. et al. *Tetrahedron*, **2015**, *71*, 5315-5320). We noticed that the triplet of H-3 at δ 4.37 remained stationary, suggesting that the activation of the thiophenyl did not disturb the boron-masked hydroxyl group. 1D-TOCSY overlay of **4a**, **4a.1** and **4a.2** was shown in Figure 2. ^{19}F and $^{13}\text{C}\{^1\text{H}\}$ spectra were shown in Figure 3 and 4, respectively. ^{11}B NMR was found to be similar to our other donors and remained stationary.

Figure 2. 1D-TOCSY overlay of **4a**, **4a.1** and **4a.2** at $-65\text{ }^{\circ}\text{C}$.

Figure 3. ^{19}F NMR after addition of Tf_2O .

Figure 4. $^{13}\text{C}\{^1\text{H}\}$ after addition of Tf_2O .

Finally, butanol was added and the ^1H anomeric triflate at δ 6.21 and ^{19}F at δ 0.12 disappeared immediately. We were unable to find suitable chemical shift resonance to directly measure the alpha:beta ratio. Thus, the reaction mixture in the NMR tube was filtered and purified by flash column chromatography to reveal an alpha:beta ratio at approximately 1:17. Given the multiple instances of raising the NMR tube to add in more reagents, we believe the stereoselectivity outcome matches well with the beta-only selectivity we observed under ideal reaction condition.

The authors responded to the reviewer's comment (7. Although glucosyl donor 12a was converted into 1,6-anhydroglucoside 12e (Table 2, entry 12), I could not understand whether 6-OH was not masked by Bu_2BOTf or alcohol exchange reaction with BuOH , followed intramolecular reaction took place rapidly.) as follows:

“In one of the NMR studies, one equivalent of BuOH was added 5 mins after introduction of one equivalent of Bu_2BOTf . We observed no change in chemical shift of boron-masked 2.2a after 30 mins at -40 °C. We concluded that the exchange process between different alcohol species was slow enough at low temperature to warrant the desired masking effect.”

However, since I am asking about the reaction of donor **12a** not donor **2.2a**, they should answer appropriately.

→ As requested, we have conducted the low temperature NMR experiments with donor **12a**.

Upon addition of Bu_2BOTf at $-40\text{ }^\circ\text{C}$, the proton of OH at C-6 disappeared and H-6 were shifted downfield to 4.21ppm from their previous positions at 3.72 ppm ($\Delta=0.49\text{ ppm}$). The comparison between **12a** at $-40\text{ }^\circ\text{C}$ and the boron-masked **12a.1** was shown in Figure 5.

Figure 5. $^1\text{H-NMR}$ of donor **12a** and the boron-masked **12a.1** at $-40\text{ }^\circ\text{C}$.

Next, butanol was added into the reaction mixture and $^1\text{H-NMR}$ was collected. As shown in Figure 6, we can observe the presence of OH proton from butanol and the still absence of OH

proton from **12a.1**. This was confirmed by 1D-TOCSY experiments (Figure 7). We monitor the reaction mixture for 45 minutes and no change was detected.

Figure 6. ^1H -NMR of boron-masked **12a.1** and butanol at $-40\text{ }^\circ\text{C}$.

Figure 7. 1D-TOCSY of boron-masked **12a.1** and butanol at $-40\text{ }^\circ\text{C}$.

3. I recommend the authors to cite relevant papers of the regio- and stereoselective glycosylation using a organoboron reagent as a transient masking group, such as *Tetrahedron Lett.* 2010, 51, 1570.

→ We thank the reviewer for these references. We have added the following papers to the manuscript:

30. Kaji, E., Nishino, T., Ishige, K., Ohya, Y. & Shirai, Y. Regioselective glycosylation of fully unprotected methyl hexopyranosides by means of transient masking of hydroxy groups with arylboronic acids. *Tetrahedron Lett.* 51, 1570-1573 (2010).
31. Kaji, E., Yamamoto, D., Shirai, Y., Ishige, K., Arai, Y., Shirahata, T., Makino, K. & Nishino, T. Thermodynamically Controlled Regioselective Glycosylation of Fully Unprotected Sugars through Bis(boronate) Intermediates. *Eur. J. Org. Chem.* 17, 3536-3539 (2014).
32. Rocheleau, S., Pottel, J., Huskić, I. & Moitessier, N. Highly Regioselective Monoacylation of Unprotected Glucopyranoside Using Transient Directing-Protecting Groups. *Eur. J. Org. Chem.* 3, 646-656 (2017).

REVIEWERS' COMMENTS:

Reviewer #2 (Remarks to the Author):

Since the authors responded the reviewer's comments (including additional experiments) point by point, now I recommend publication of this manuscript in Nature Communications.